# Optimization of the Extraction of Antioxidant Compounds from Roselle Hibiscus Calyxes (*Hibiscus sabdariffa*), as a Source of Nutraceutical Beverages

**DOI:** 10.3390/molecules28062628

**Published:** 2023-03-14

**Authors:** María José Villalobos-Vega, Gerardo Rodríguez-Rodríguez, Orlando Armijo-Montes, Pablo Jiménez-Bonilla, Víctor Álvarez-Valverde

**Affiliations:** 1Laboratorio de Fitoquímica (LAFIT), Escuela de Química, Universidad Nacional (UNA), Heredia 40101, Costa Ricapablo.jimenez.bonilla@una.ac.cr (P.J.-B.); 2Escuela de Química, Universidad Nacional (UNA), Sede Interuniversitaria de Alajuela, Heredia 40101, Costa Rica; 3Laboratorio de Productos Naturales y Ensayos Biológicos (LAPRONEB), Escuela de Química, Universidad Nacional (UNA), Heredia 40101, Costa Rica

**Keywords:** *Hibiscus sabdariffa*, antioxidants, nutraceuticals, phenolic compounds, functional foods

## Abstract

Secondary metabolites from *Hibiscus sabdariffa* have been used to prevent different diseases. Roselle Hibiscus is known for being rich in phenolic bioactive compounds. The extraction conditions are directly related to the chemical composition and then to the overall bioactivity of the extract. In this study, a Box-Behnken experimental design has been used to optimize the antioxidant activity, considering four variables: ethanol:water ratio, temperature, extraction time, and solvent:solid ratio. The experiment comprises 27 experiments and 3 repetitions at the central point. The results are described by surface response analysis and a second-degree polynomial equation. The model explains 87% of the variation in the response. The maximum antioxidant activity is yielded when 1% solids are extracted in 35.5% ethanol at 60 °C for 33 min. Finally, a nutritional functional supplement of 495 µmol Trolox Equivalent (TE) antioxidant capacity was prepared with the optimized extract.

## 1. Introduction

Free radicals (FR) are being constantly generated in living organisms. FR are byproducts of some metabolic processes; examples include hydroxide, peroxide, and superoxide radicals. Oxidative stress is produced when FR levels exceed the threshold limit due to physiological reasons [1], and it damages biomolecules such as lipids, nucleic acids, carbohydrates, and proteins. Oxidative stress on cells and tissues causes aging in humans. Several oxidizing agents are produced by endogenous and exogenous processes; these agents are called Reactive Oxygen Species (ROS). The constant oxidative stress on living organisms has an impact on cellular and physiological processes. The living cells decrease their damage by eliminating FR through their own mechanisms. However, the endogenous mechanisms start losing efficiency due to aging and other physiological processes; an exogenous supplement is then required [2]. Oxidative stress and aging are related to several diseases, such as diabetes, cancer, cardiovascular illnesses, and others [3]. Antioxidant compounds eliminate the free radicals of ROS through hydrogen atom transfer (HAT), single electron transfer (SET), or chelation using transition metals [4]. These compounds can be sorted into two classes: radicals and non-radicals.

Phenolic compounds, carotenoids, and vitamins are some metabolites from fruits and vegetables responsible for antioxidant capacity [5]. Roselle Hibiscus is a tropical plant also known as rozelle, sorrel, red sorrel, Jamaican sorrel, Indian sorrel, Guinea sorrel, sour-sour, queensland jelly plant, jelly okra, lemon bush, and Florida cranberry [6]. *H. Sabdariffa* belongs to the Malvaceae family. It grows in bushes (annual) with solitary flowers, and the calyxes are fleshy and red. It is easy to culture all over the world, and the food and pharmaceutical industry utilizes its nutraceutical properties. Plants grow deep and require sandy ground with potassium and organic matter; they also require sun and a pH between 4.5 and 8.0 [7]. Two main varieties of *H. sabdariffa* are known: var. *altissima Wester* (fiber-producing plant), and var. *sabdariffa* (food plant) [8]. *Hibiscus sabdariffa* calyx extract is a rich source of anthocyanin as a potential antioxidant [9], with antioxidant efficacy due to the presence of cyanidin-3-glucoside, delphinidin-3-glucoside, cyanidin-3-sambubioside, and delphinidin-3-sambubioside. These anthocyanins are responsible for the characteristic red color of the H. sabdariffa calyxes and can be recovered for subsequent use as colorants in different industrial sectors.

Water-based roselle calyx extracts are used worldwide to prepare good-tasting beverages with antioxidant properties [10]. The antioxidant properties are related to the content of organic acids, flavonoids, and phenolic acids [11,12,13]. Roselle extracts have functional properties and are of interest for nutraceutical product development. These bioactive products can reduce the risk for some illnesses and improve some organ functions and overall health. Roselle is cytocompatible and it can even replace dyes for histological staining [14].

Previous studies have evaluated different solvents [15], including water [16], for calyx extraction. Also, different solid/solvent ratios are reported, from 1–10 [17,18] to 100 or greater [19]. Most previous articles have focused on extraction time and stability of molecules [20], anthocyanin yield [21], or specific bioactivities such as enzymatic inhibition [22]. In this work, we optimize the antioxidant capacity using a Box-Behnken design. This information is useful for reducing costs and time, and to obtain more functional foods. This work aims to optimize the antioxidant activity of extracts from *H. sabdariffa* calyxes for use in nutraceutical products.

## 2. Results and Discussion

Solid-liquid extraction is a separation process used for transferring solutes from a solid matrix to a solvent. This technique is used to obtain bioactive compounds from plants. The efficiency of solid-liquid extraction is related to many factors such as temperature, solvent composition, stirring speed, solid-liquid rate, time, particle size, pH, and others. Four variables shown in Table 1 were selected based on a bibliographic review [23,24,25,26,27].

BBD was utilized to find the best extraction conditions for Roselle Hibiscus to optimize the antioxidant capacity. The experimental values obtained in this work were used to obtain the second-order empirical coefficients for each variable. Only significant coefficients (*p*-value < 0.05) for both the variables and the interactions were included in the model. Equation (1) describes the overall polynomial model explaining the antioxidant capacity in terms of ethanol:water, temperature (°C), time (min), and solid/solvent ratio, described as X_1_, X_2_, X_3_, and X_4_, respectively.
µmol TE/gDM = 94.72 − 18.95 X_1_ + 9.03 X_2_ − 15.63 X_4_ − 35.16 X_1_X_1_ − 15.43 X_2_X_2_ − 10.00 X_3_X_3_ + 7.35 X_1_X_2_(1)

Results of ANOVA test can be found in Table 2; *p*-value and F-value for the regression (Equation (1)) were <0.0001 and 36.190, respectively.

Analysis of variance was realized for the adjusted model for the antioxidant capacity of hibiscus extracts. The determination coefficient (R^2^) of the model is 0.940. The R^2^ value confirms that the regression model explains well the actual behavior of the system [28]. The adjusted R^2^ is 0.9241. Although it is smaller than the regular determination coefficient, both values are very close to each other; this means the values predicted by the model are a good representation of the experimental results [29].

A lack-of-fit test showed a *p*-value of 0.108. It is higher than 0.05, meaning there is no evidence of a lack of fit. The model is an appropriate representation of the relationship between the experimental factors and the response variable [28].

The regression model predicts the effect of the four variables on the antioxidant capacity after the extraction process. The relationship between dependent and independent variables is illustrated through the surface 3D graphs generated from the model (Figure 1). The optimal points from the 3D graphs are the highest antioxidant capacity from the subset of conditions considered within the graph.

### 2.1. Effect of Ethanol Content on Antioxidant Capacity

Figure 1A–C show the effect of increasing the ethanol content of the solvent on the extraction for the solid/solvent ratio, temperature, and time, respectively. In each of Figure 1A–C, the extraction efficiency of antioxidants starts increasing when the ethanol concentration in the solvent increases from 0 to 34.5% ethanol. However, the antioxidant extraction efficiency decreases when the ethanol concentration increases by more than 34.5% in the extraction solvent. When the ethanol:water ratio is fixed at 34.5:65.5, the maximum antioxidant capacity is reached at a low solid/solvent ratio, intermediate temperature values, and intermediate time values.

Low antioxidant capacity of the extracts is obtained when the ethanol concentration is high in the extractant phase [30]. These results may confirm the high efficiency of water:ethanol as a solvent for evaluating the antioxidant capacity of *H. Sabdariffa*. Antioxidants found in literature for hibiscus aqueous-ethanolic extracts are organic acids, phenolic acids, flavonoids, and anthocyanins [19]. These compounds are soluble in hydroalcoholic mixtures containing equal amounts of ethanol and water, but their solubility decreases when ethanol concentration is near the azeotrope. In our results, the best antioxidant capacity is obtained when the solvent is composed of a mixture of water and organic solvent (Table 1). The combination of these solvents dissolves a wide range of phenolic compounds [31]. The dipoles from phenolic compounds (such as delphinidin-3-*O*-sambubioside and cyanidine-3-*O*-sambubioside) interact with the dipoles from ethanol and water, yielding a higher extraction rate [32].

The optimal ethanol concentration in the solvent is 34.5%. There is no additional improvement in antioxidant capacity when the ethanol concentration is increased above 34.5%. This behavior is due to the average affinity of the antioxidant compounds. Nonetheless, a higher water concentration could promote the degradation of anthocyanidins [33] because of Flavylium cation stability. This ion is the predominant form of anthocyanins in acidic medium. Flavylium ion is susceptible to the nucleophilic attack of water, and after reacting it generates a pseudo hemiketal with reduced antioxidant capacity [34].

### 2.2. Effect of Extraction Temperature on Antioxidant Capacity

Temperature is an important parameter in the extraction of hibiscus. Figure 1B,E,F show the effect of temperature, solid-solvent ratio, and time on the antioxidant capacity of the extract. The highest antioxidant capacity is reached at 60 °C, combined with low solid/solvent ratios and medium values for % ethanol in solvent and time. Similar behavior has been reported [35]. They found that the total phenolics and the antioxidant capacity of the extract decayed when the temperature was increased to 90 °C. High temperatures can decompose or modify thermosensitive bioactive compounds such as polyphenols and other antioxidants [36]. Also, very high temperatures can accelerate the co-extraction of other non-active components such as sugars and fiber; the antioxidant concentration in the extract would then decrease [37].

A low concentration is also inconvenient for obtaining an antioxidant-rich extract. The cell wall from hibiscus is weakened when the temperature is increased. As a result, the cellular components and the chemical compounds have more interaction with the solvents. [38]. Results from Figure 1B,E,F show that the best temperature for extraction of antioxidants from hibiscus is 60 °C. Commonly, hibiscus calyxes are boiled to obtain infusions. However, our results suggest hibiscus should not be boiled, as that process decreases its quality.

### 2.3. Effect of Extraction Time on Antioxidant Capacity

Figure 1C,D,F show the effect of time on antioxidant capacity, with respect to the % ethanol in the solvent (Figure 1C), the solid/solvent ratio (Figure 1D), and the temperature (Figure 1F). There are no significant differences in antioxidant activity when the time changes between the limits included in this study. However, prolonged extraction times at high temperatures can decompose and oxidize phenolic compounds; consequently, the antioxidant capacity is reduced [39].

The extract with the best antioxidant capacity was obtained when the extraction temperature was kept at 60 °C for 33 min. According to Ramírez et al. [25], the concentration of polyphenolic compounds increases with the extraction time, at the appropriate temperature. The compounds need time to migrate to the solvent [39].

### 2.4. Effect of Solid/Solvent Ratio on Antioxidant Capacity

The solid/solvent ratio is an important parameter during the extraction process. The solid is the mass of powdered calyxes from H. sabdariffa, and the solvent is the hydroalcoholic mixture. Figure 1A,D,E show the surface response graph for 500 mL total volume and variable solid/solvent ratio. For example, 5 g of hibiscus extracted with 500 mL of solvent represents a 1/100 ratio. The maximum antioxidant capacity was reached at this condition (1/100 solid/solvent). The antioxidant activity increases with decreasing solid/solvent ratio until it reaches the optimal value. Similar results were seen by Tan et al. [40]. A greater concentration gradient increases the diffusion rate when the solvent amount increases; this is consistent with the principle of mass transfer [41]. This trend can be reversed at any point by the dilution rate. However, this behavior is not observed within the conditions included in this study.

### 2.5. Contribution of Total Polyphenolic Compounds (TPC) and Anthocyanins (ACs) to Antioxidant Activity

The two main ACs (delphinidin-3-*O*-sambubioside and cyanidine-3-*O*-sambubioside), and the TPC were determined for each extract. The ACs selected were found as main components of hibiscus aqueous extracts by Segura et al. [42].

According to Ramírez et al. [25], ACs handle 51% of antioxidant capacity. Figure 2 shows TPC and ACs for the 26 experiments included in the BBD. TPC values are higher than ACs as expected. Although both TPC and ACs follow the same trend, both are related to antioxidant capacity. Also, the experiment runs showing lower content of both TPC and ACs were extracted with 95% ethanol; this means that ethanol is not a good solvent for extracting phenolic compounds.

### 2.6. Study of a Formula Prepared Using Optimized Hibiscus Extract as a Base Component

Nutraceutical products can prevent or treat several diseases. Nutraceutical intake is recommended as complementary or alternative treatments [43]. Primarily, hibiscus is commercially utilized to prepare hot beverages. For this reason, in this work, a bioactive nutraceutical product was formulated using hibiscus extract. When the extract was obtained using optimized conditions (as shown in Figure 3), it showed an antioxidant capacity of 103.36 µmol TE/g DM, with a prediction error of 10.71% when compared to the prediction value obtained in the statistical analysis (115.76 µmol TE/g DM; shown in Figure 3). The hibiscus dried extract was dried by atomization and blended with a non-caloric sweetener and polydextrose. This procedure helps to preserve the stability of the bioactive compounds and requires less time than freeze-drying or vacuum-drying [44]. This drying process has some advantages: the powder is lighter, its volume is decreased, and it is easier to handle and transport. The spray-drying technique produces a microbiological- and oxidative-resistant powder. The method is simple, easily automatable, and fast. However, the atomization drying yield is 30% lower than freeze- drying, but the process can still be improved by optimizing parameters such as inlet temperature, feeding flux, and outlet temperature. The proximate analysis of the resulting powder shows 1.14 µg/g of total sugars, 3.81 Kcal/g, pH 3.30, and 0.258 mg/g of sodium. There is not a standard daily consumption of antioxidants; however, the United States Department of Agriculture (USDA) recommends ingestion of 3000–5000 µmol TE [45].

A mass of 5 g of the previously prepared formula can be dissolved in 100 mL of water. The drink showed an antioxidant capacity of 495 µmol TE. Our antioxidant activities are close to those shown by other reference herbal infusions such as green tea [46]. However, most ready-to-drink formulas are supplemented with ascorbic acid and other antioxidant compounds [47]. The formula proposed in this study is only supplemented with stevia as a sweetener and polydextrose for flavor stabilization. The only source of antioxidants in the proposed formulation is the herbal extract.

Some of the advantages of a dry powder formula are that the antioxidant activity is not affected by preparation conditions [47], the humidity content is very low, the product is stable for a long time [48], and the phytochemical characteristics are preserved.

## 3. Materials and Methods

### 3.1. Plant Material Preparation and Maceration

Dry whole roselle calyxes were purchased from Doña Rosa Food Products (San Isidro, Heredia, Costa Rica). The material was freeze-dried in a Freezone 2.5 Plus (from Labconco Corp., Kansas City, MO, USA), and ground to 2 mm in a cutting mill SM100 (from Restsch GmbH, Haan, Germany). Samples were stored at room temperature.

A double-jacket laboratory reactor model LR-2.ST (from IKA WERKE, Staufen, Germany) was equipped with a circulating thermostat Ecoline E306 (from LAUDA, Lauda-Königshofen, Germany) and was used for the extraction procedure. The reaction flask was equipped with a glass condenser and deflectors. Mechanical agitation was kept at 100 rpm. Different ratios of previously ground material and solvent were tested. The reactor was filled with 500 mL of the selected solvent and the corresponding previously ground roselle calyxes. Specific extraction times, temperature, solvent composition, and solid/solvent ratio are explained in the experimental design section. The extracts were used directly for further analyses.

### 3.2. Antioxidant Activity Determination

The antioxidant activity was measured by the 2,2-diphenyl-1-picrylhydrazyl (DPPH) antiradical test. We followed the 96-well microplate protocol [49]; 30 µL of each extracted sample was mixed with 270 µL of a 0,04 mg/mL DPPH solution in 80% methanol. After 20 min of incubation at room temperature, absorbance was measured at 515 nm in a Synergy HT Multi-Mode microplate reader (from BioTek Instruments, EUA, Winooski, VT, USA). A standard curve ranging from 0 to 250 µmol/mL 6-hydroxy-2,5,7,8-tetramethylchroman-2-carboxylic acid (Trolox) was used. Sample antioxidant activity was reported as Trolox Equivalents per gram of Dry Mass (µmol TE/DM).

The ORAC procedure used a Synergy HT Multi-Mode automated plate reader (from BioTek Instruments, EUA, Winooski, VT, USA) [50]. Analyses were conducted in phosphate buffer pH 7,4. Peroxyl radical was generated using 2,2-azobis(2-amidino-propane) dihydrochloride, which was prepared fresh for each run. The standard curve was linear between 0 and 125 µmol/L Trolox. Fluorescein was used as the substrate. Fluorescence conditions were as follows: excitation at 485 nm and emission at 520 nm. Sample antioxidant activity was reported as Trolox Equivalents per gram of Dry Mass (µmol TE/DM).

### 3.3. Box-Behnken Design (BBD)

Box-Behnken design was used; it comprises 27 experiments with 3 central points, and 4 variables at three levels. Variables were as follows: solvent composition (ethanol:water ratio), temperature, time, and solid/solvent ratio (ground roselle mass and volume of ethanol/water). Table 3 explains the levels for each variable selected. The response variable is the Antioxidant Activity determined by the DPPH method.

### 3.4. Total Phenolic Content (TPC) Determination

TPC was determined by employing colorimetry using the Folin-Ciocalteu method. We followed the 96-well microplate procedure developed by Sánchez-Rangel et al. (2013) [51], with minor modifications. 30 µL of the sample was mixed in a well with 200 µL of distilled water, 15 µL of Folin-Ciocalteu reagent, and 50 µL of 20% Na_2_CO_3_ solution. Then, the mixture was incubated for 20 min while mixing in a Synergy HT Multi-Detection Microplate Reader (BioTek Instruments, Winooski, VT, USA) at 40 °C. Finally, absorbance was measured at 755 nm against a standard curve of 0.000, 0.020, 0.040, 0.060, 0.080, and 0.120 mg/1 mL of gallic acid.

### 3.5. Identification and Quantification of Anthocyanins

Two anthocyanins (delphinidin-3-*O*-sambubioside and cyanidin-3-*O*-sambubioside) were quantified using HPLC-DAD Ultimate 3000 (from Term Scientific Instruments, EUA, West Palm Beach, FL, USA). These compounds were previously isolated from roselle and characterized by Segura et al. [42]. The separation was performed using a C18 Dionex Acclaim (250 × 4.0 mm, 5 µm) column. A constant solvent flux of 0.8 mL/min was utilized. A binary pump filled with aqueous 0.01% TFA (trifluoroacetic acid) and acetonitrile as mobile phase components were used. Initially, the pump was set to keep acetonitrile constant at 10% from time 0 to 8 min, then it was linearly increased to 50% (10 min) and finally increased to 95% (13 min). The diluted TFA completes the remaining sambubioside and cyanidin-3-*O*-sambubioside (from Sigma-Aldrich, St. Louis, MO, USA) as standard, and read at 530 nm. Chromatographs are shown in Appendix A.

### 3.6. Preparation and Characterization of a Nutraceutical Drink Using Optimized Roselle Extract as Raw Material

The optimized roselle extract was concentrated in a rotatory evaporator B-490, then the concentrated extract was dried out in a Mini Spray Dryer B-290 (both from Büchi Corporation, Flawil, Switzerland). Finally, the powder of the highest-antioxidant extract was supplemented with stevia (as sweetener) and polydextrose (vehicle) in a mass ratio of 1:1:3 (extract:stevia:polydextrose).

The caloric content of the final product was determined using a bomb calorimeter model C 200 (from IKA WERKE, Staufen, Germany), according to the protocol DIN 51900-1 (DIN, 2000) [52]. Total carbohydrates were determined by the colorimetric phenol-sulfuric acid method [53]. Sodium was determined by atomic absorption following the AOAC 963.09 procedure [54].

### 3.7. Statistical Analysis

All samples were quantified in duplicate and the results were expressed as mean ± standard deviation. Data processing and statistical analysis (mean value) were performed using Microsoft Excel 2019. Response surface design and statistical analysis of the model were performed through an ANOVA; Minitab version 19 was used for these purposes.

## 4. Conclusions

*H. sabdariffa* calyxes are natural products with functional properties such as high antioxidant levels. Products prepared from these calyxes can protect humans from different health disorders—the diverse range of potential products is a good reason for taking advantage of their functional properties.

A viable method for extraction of *H. sabdariffa* calyxes was developed after optimization of multivariate experimental conditions. The optimal extraction conditions for antioxidant capacity were determined using a Box-Behnken experimental design.

The extraction method decreased the extraction time to 33 min, using 5 g of sample and 500 mL of 34.5% ethanol as solvent, at 60 °C. The method employs a reduced concentration of organic solvent and uses a renewable organic solvent. Thus, this method can be considered environmentally friendly.

Ethanol concentration in the solvent is considered the most important variable for the extraction of hibiscus calyxes. The optimal ethanol concentration was found to be 34.5%. No significant improvement is observed in the extraction when ethanol concentration is increased above that value. The reason for this is the basic principle of extraction, known as affinity.

In general terms, the extracts exhibiting a higher ethanol concentration also presented a low TPC until the optimum value was reached. This effect affects the antioxidant activity, for which a similar trend was observed.

TPC and PAC from Roselle Hibiscus extract are strictly related and both contribute to antioxidant capacity.

Nutraceutical beverages can promote a healthy life and prevent diseases; these beverages are functional and low-calorie alternatives.

## Figures and Tables

**Figure 1 molecules-28-02628-f001:**
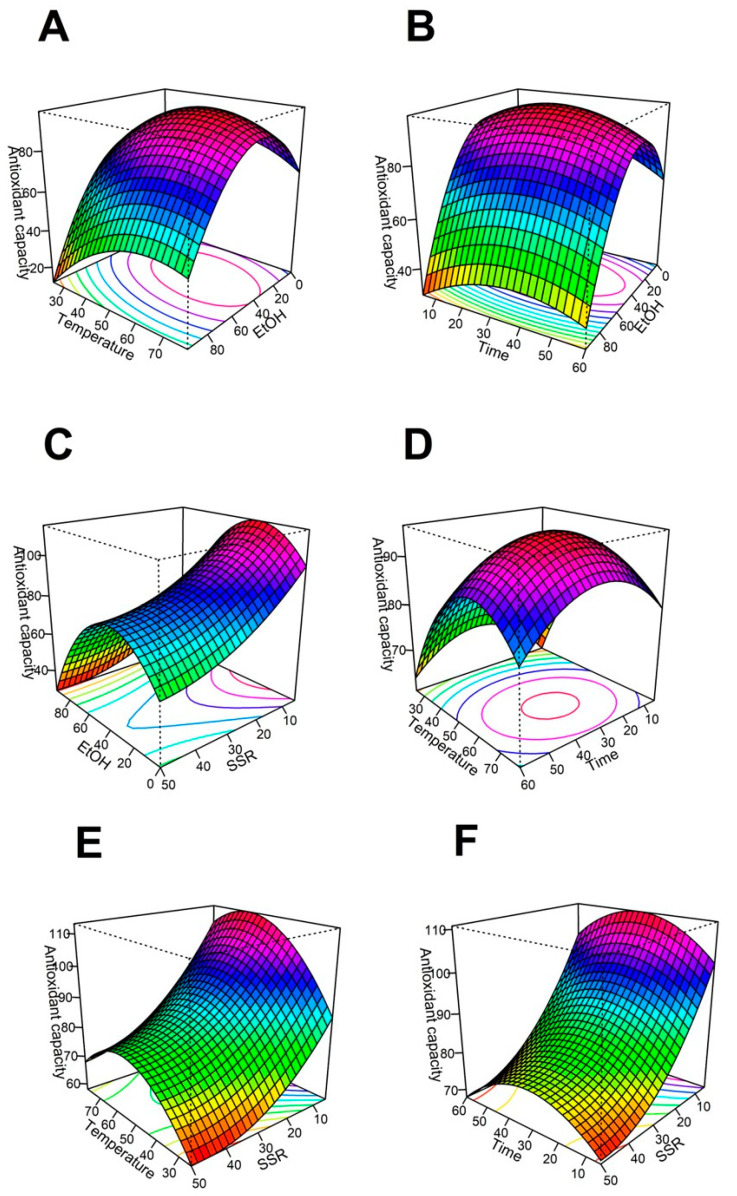
Surface-response analysis of the effect of EtOH (%), Temperature (°C), Time (min), and SSR (g/500 mL) on the antioxidant capacity (µmol TE/gDM). (**A**) SSR vs. EtOH, (**B**) EtOH vs. temperature, (**C**) EtOH vs. Time, (**D**) Time vs. SSR, (**E**) Temperature vs. SSR, and (**F**) Temperature vs. Time. Abbreviations: TE: Trolox equivalents, DM: dry mass, SSR: solid/solvent ratio, EtOH (mass % ethanol in solvent; the remaining percent corresponds to water).

**Figure 2 molecules-28-02628-f002:**
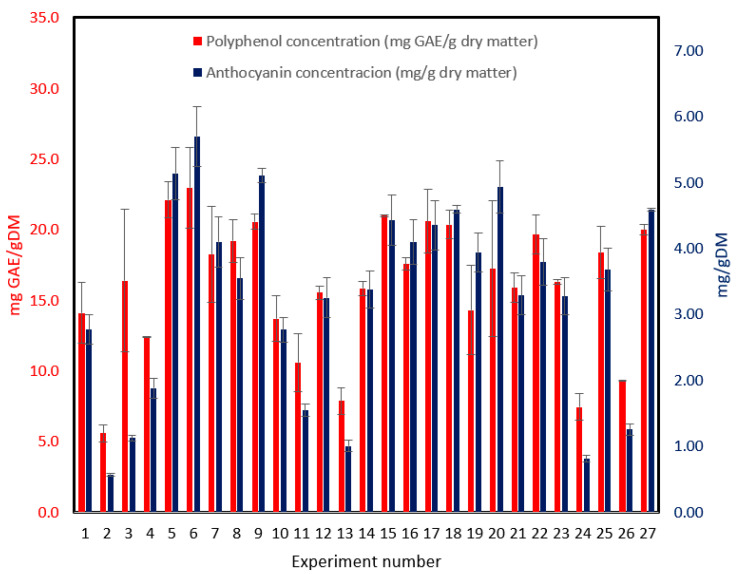
Total polyphenolic compounds and anthocyanins for individual extraction experiments from *H. sabdariffa*. Error bars represent standard deviation. Tuckey test results for the 27 experiments are shown in Appendix A (TPC) and Appendix A (PAC).

**Figure 3 molecules-28-02628-f003:**
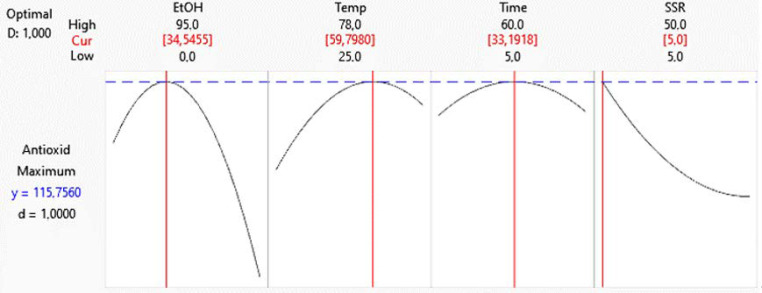
Optimized variables of the extraction of antioxidant compounds from *Hibiscus sabdariffa* calyxes.

**Table 1 molecules-28-02628-t001:** Data matrix of Box-Behnken design for extraction of antioxidants from hibiscus calyxes.

Experiment Number	X_1_	X_2_	X_3_	X_4_	Antioxidant Capacity (µmol TE/ g DM ± SD)	Adjust
1	−1	−1	0	0	68.24 ± 2.46	71.97
2	1	−1	0	0	20.05 ± 1.72	19.37
3	−1	1	0	0	72.45 ± 1.73	75.34
4	1	1	0	0	53.22 ± 3.55	52.13
5	0	0	−1	−1	95.08 ± 13.37	97.17
6	0	0	1	−1	102.46 ± 1.58	97.17
7	0	0	−1	1	65.14 ± 0.17	65.91
8	0	0	1	1	65.60 ± 2.03	65.91
9	0	0	0	0	106.35 ± 25.55	103.61
10	−1	0	0	−1	114.10 ± 19.46	99.49
11	1	0	0	−1	66.33 ± 1.36	60.76
12	−1	0	0	1	58.47 ± 2.46	68.23
13	1	0	0	1	27.00 ± 3.68	29.50
14	0	−1	−1	0	73.72 ± 0.88	76.51
15	0	1	−1	0	99.23 ± 4.84	95.34
16	0	−1	1	0	75.61 ± 6.94	76.51
17	0	1	1	0	96.10 ± 3.98	95.34
18	0	0	0	0	101.81 ± 10.57	110.69
19	0	−1	0	−1	81.44 ± 3.66	76.45
20	0	1	0	−1	31.96 ± 2.72	37.72
21	0	−1	0	1	73.23 ± 5.17	76.45
22	0	1	0	1	38.62 ± 0.51	37.72
23	−1	0	−1	0	74.69 ± 2.91	80.24
24	1	0	−1	0	93.61 ± 2.41	99.08
25	−1	0	1	0	61.27 ± 2.46	48.98
26	1	0	1	0	70.43 ± 1.72	67.82
27	0	0	0	0	100.23 ± 1.73	101,04

Xn refers to the levels of the experiment, TE: Trolox equivalents, DM: Dry Mater, SD: Standard Deviation.

**Table 2 molecules-28-02628-t002:** Estimated regression coefficients, model adequacy checking, and ANOVA analysis of the model.

Source	DF	SSq Adjust	MC Adjust	F-Value	*p*-Value
Model	9	15,361.9	1706.9	36.2	<0.0001
X_1_	1	4291.9	4291.9	91.0	<0.0001
X_2_	1	975.3	975.3	20.7	<0.0001
X_4_	1	3719.4	3719.4	78.9	<0.0001
X_1_X_1_	1	7174.4	7174.4	152.1	<0.0001
X_2_X_2_	1	1381.3	1381.3	29.3	<0.0001
X_3_X_3_	1	574.6	574.5	12.2	0.003
X_1_X_2_	1	216.7	216.7	4.6	0.047
Lack of fit	16	1400.0	87.5	8.7	0.108
	R-Squared		0.9504		
	Adjusted R-Squared		0.9241		
	Predicted R-Squared		0.8747		

DF: Degrees of Freedom. SSq: Sum of Square, MC: Model Checking.

**Table 3 molecules-28-02628-t003:** Independent variables and levels for Box-Behnken experimental design.

Factor	Symbol	Level		
		−1	0	1
Ethanol:water	X_1_	5:95	50:50	0:100
Temperature (°C)	X_2_	25	50	78
Time (min)	X_3_	5	30	60
Solid/Solvent ratio	X_4_	1/100	1/50	1/10

Xn refers to the symbol used to describe the used level combination.

## Data Availability

Data supporting reported results are available upon request; please contact victor.alvarez.valverde@una.ac.cr.

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
