# Peer review of "Optimization of the Extraction of Antioxidant Compounds from Roselle Hibiscus Calyxes (Hibiscus sabdariffa), as a Source of Nutraceutical Beverages"

_molecules, 2023, doi:10.3390/molecules28062628_

Round 1

Reviewer 1 Report (Previous Reviewer 1)

The study is interesting, require bit modification with addition of suggested results. 

1.Line 34-35: Phenolic compounds, carotenoids, and vitamins are some metabolites from fruits and vegetables responsible for antioxidant capacity. Hibiscus sabdariffa Calyx extract have also indicated rich source of anthocyanin as potential antioxidant (https://doi.org/10.1007/s10924-022-02596-x) with antioxidant efficacy due to presence of cyanidin-3-glucoside, delphinidin-3-glucoside, cyanidin-3-sambubioside, and delphinidin-3-sambubioside.

2. Suggested to report the total flavonoid contents with phenolic content

3. Suggested to report the radical scavenging efficacy of optimised formulation following DPPH and FRAP

4. Application of Hibiscus sabdariffa Calyx as nutraceutical drink, require stability study results demonstration, which is an important pre-requist of ready to use energy drink

5. WIth Calorie  evaluations of drink further assessment of drinks are required such as pH, viscosity, etc. related to drinks

6.  Although previous several study reported excellent antimicrobial efficacy Hibiscus sabdariffa, suggested to test the drink after storage in various condition for microbial growth since sweetener and other additives were added in the formulation.

Good Luck

Author Response

Reviewer 1

Comment: There are several comments from a reviewer involving an important amount of additional lab work.  We decided to do not include additional experiments at this point, considering the editor let us 10 days to answer the comments, and our objectives were achieved with our previous experimentation. Besides the reviewers comments, we had included some paragraphs in introduction and conclusions in order to accomplish with the 4000 words (minimum recommended)

Line 34-35: Phenolic compounds, carotenoids, and vitamins are some metabolites from fruits and vegetables responsible for antioxidant capacity. Hibiscus sabdariffa Calyx extract have also indicated rich source of anthocyanin as potential antioxidant (https://doi.org/10.1007/s10924-022-02596-x) with antioxidant efficacy due to presence of cyanidin-3-glucoside, delphinidin-3-glucoside, cyanidin-3-sambubioside, and delphinidin-3-sambubioside.

R/ Done

  1. Suggested to report the total flavonoid contents with phenolic content

R/ We agreed this is a good suggestion, but we decided to report proanthocyanidin, because flavonoids are the most important subgroup present in H. Sabdariffa PAC’s. 51% of the antioxidant activity of the calyxes correspond to PACs (https://doi.org/10.1016/S0963-9969(01)00129-6). Also, the TPC analysis includes flavonoids with a wider perspective of the antioxidant compounds.

  1. Suggested to report the radical scavenging efficacy of optimised formulation following DPPH and FRAP

R/ Again we considered this is a good suggestion, but we considered the information is not required to obtain our conclusions. The optimization of the extraction process of H. sabdarifa had utilized DPPH method for antioxidant capacity quantification. We also included ORAC for the optimized extract characterization, to better compare it with other previous reports. We know the reviewer suggested reporting antioxidant capacity with different methods. We know that information will enrich the final characterization, but it will take a long time at this point, and most likely is not going to change anything of our conclusions.

  1. Application of Hibiscus sabdariffa Calyx as nutraceutical drink, require stability study results demonstration, which is an important pre-requist of ready to use energy drink

R/ Our main objective was the optimization of the antioxidant capacity of Hibiscus. Additionally, we showed a potential application, for preparing a ready-to-use drink.  Also, since the formulation is a powder, stability is probably not a big deal.

  1. WIth Calorie  evaluations of drink further assessment of drinks are required such as pH, viscosity, etc. related to drinks.

R/ pH value was included in section 2.6. All the additional recommendations are a great suggestion, but we consider these can be considered in future investigations. Those results are not necessary to  achieve our objectives.

  1. Although previous several study reported excellent antimicrobial efficacy Hibiscus sabdariffa, suggested to test the drink after storage in various condition for microbial growth since sweetener and other additives were added in the formulation

R/ Again we consider this suggestion is very good although, we consider this go moreover the scope of this research.

Reviewer 2 Report (Previous Reviewer 3)

Accept

Author Response

Thank you for your coment.

Round 2

Reviewer 1 Report (Previous Reviewer 1)

At present stage authors have taken decision by himself. So he/she can decide on acceptance of the manuscript. Review process are done to improve the manuscript. Stability of solid are not big deal what about drink prepared using the that solid?. Thanks and good luck 

This manuscript is a resubmission of an earlier submission. The following is a list of the peer review reports and author responses from that submission.

Round 1

Reviewer 1 Report

Although authors reported a well planned work, some improvements are required following below suggested suggestions.

1. Suggested to report recent data on roselle calyxes supported with extraction values in introduction to justify the aim and objective of the research, also explain how the optimisation technique is more effective than conventional extraction where generally solid to solvent ration kept twice or thrice.

2. What was the particle size of previously-ground roselle calyxes powder or mesh size through which it was passed to reduce the size.

3. What was the hypothesis of testing against only DPPH, Although several methods are available to test scavenging efficacy generally radical scavenging efficacy are tested for DPPH together with ABTS and FRAP. Suggested to test the optimised condition extract with other reagent too.

4. The figure quality of Surface-response analysis is low suggested to add at least 300 DPI images.

5. Suggested to elaborate the obtained results, figure cannot explain themselves or else combine results and discussion sections to the best

6. Suggested to add statistical significance in figure 2

7.  Please elaborate on this "The water and the flavylium cation can form a hemiketal pseudo base, and then the base can produce further reactions."

8. In temperature dependent extraction, did you tried to check the stability of tested anthocyanin, whether it was stable at optimised condition

9. Suggested to report the ready to use roselle calyxes product as drink against other than DPPH

10.  Suggested to report the cyto-compatability of ready to use roselle calyxes product against suitable cell line, or support with previous reports.

11. Few typological errors are there in the draft as indicated in the attachment. 

Goodluck for revisions.

Reviewer 2 Report

The manuscript entitled "Optimization of the extraction of antioxidant compounds from Roselle Hibiscus calyxes (Hibiscus sabdariffa), as a source of nutraceutical beverages" reports the application of a 4 parameter Box-Behnken design to extract anthocyanins and phenolic compounds from Hibiscus sabdariffa. However, there are several issues that the authors did not address properly:

-Introduction - There is no mention of previous works performed to extract bioactive compounds from this species. Authors could have mentioned, for instance, https://doi.org/10.1016/S0963-9969(01)00129-6; doi:10.1016/j.jfoodeng.2011.10.012; https://doi.org/10.3109/01480545.2010.536767. Moreover, in line 31, the authors could have explained that there are two kinds of reactive oxygen species, namely, radicals and non-radicals. In line 39, the authors could have specified which functional properties are attributed to the species under study.

Results - There is no text explaining the results and citing figures and tables.

-Table 1 - Is "adjust" the predicted values retrieved by the model? 

-Table 2 - p-values equal to 0.000 are not valid. Please, state the actual value or put <0.0001 if it is the case.

-Figure 2 -  Where are the results of the 3 central points (runs 9, 18 and 27)?  TPC values should be reported as mg GAE/g dry matter instead of mg AG/g dry matter.

Line 80 - Delete "y".

Line 86 - Analysis of variance instead of Variance analysis.

Line 108 - The sentence does not make sense.

Lines 115 and 174 - delphinidin-3-O-sambubioside instead of definidine-3-O-sambubioside.

Line 178 - 26 or 27 runs? There are 3 runs missing in Figure 2.

Table 3 - In the Box-Behnken design, the levels are equidistant but in this study, the authors did not obey this rule.

Supplementary material - The authors have provided the chromatograms of 2 anthocyanins standards. It would have been more useful to provide a representative chromatogram of one of the 27 runs. 

For all these reasons, I recommend that this manuscript should be rejected.

Reviewer 3 Report

1. Author must give the desirability graph and value of the optimization. 

2. Why the author has not applied 100 % ethanol.

3. What is the standard compound used in the antioxidant activity.

4. Apply statistics in the antioxidant activity graph 2. 

5. How much is the actuial predicted value. 

6. Calculate the prediction error.

7. Check for english n grammer error. 

8./ Discuusion must be revised with some references. 
